# Near-Infrared Fluorescence Imaging with Indocyanine Green for Robot-Assisted Partial Nephrectomy: A Systematic Review and Meta-Analysis

**DOI:** 10.3390/cancers15235560

**Published:** 2023-11-24

**Authors:** Carlo Giulioni, Prashant Motiram Mulawkar, Daniele Castellani, Virgilio De Stefano, Carlotta Nedbal, Nariman Gadzhiev, Giacomo Maria Pirola, Yu Xi Terence Law, Marcelo Langer Wroclawski, William Ong Lay Keat, Ho Yee Tiong, Bhaskar Kumar Somani, Andrea Benedetto Galosi, Vineet Gauhar

**Affiliations:** 1Urology Unit, Azienda Ospedaliero Universitaria delle Marche, Polytechnic University of Marche, 60121 Ancona, Italy; castellanidaniele@gmail.com (D.C.); virgilio.destefano@gmail.com (V.D.S.); carlottanedbal@gmail.com (C.N.); galosiab@yahoo.it (A.B.G.); 2Department of Urology, Tirthankar Super Speciality Hospital, Akola 444001, India; pmulawkar@hotmail.com; 3Government Medical College, Srinagar’s Super Speciality Hospital, 3QPX+9GP, Bemina, Srinagar 190010, India; 4Urology Unit, University of Edinburgh, Edinburgh EH8 9YL, UK; 5Department of Urology, Saint-Petersburg State University Hospital, Petersburg 190103, Russia; nariman.gadjiev@gmail.com; 6Urology Unit, San Giuseppe Hospital, Multimedica Group, 20123 Milan, Italy; gmo.pirola@gmail.com; 7Department of Urology, National University Hospital, National University Health System, Singapore 119228, Singapore; terencelaw83@gmail.com (Y.X.T.L.); surthy@nus.edu.sg (H.Y.T.); 8Hospital Israelita Albert Einstein, São Paulo 05652-900, Brazil; urologia.marcelo@gmail.com; 9Department of Urology, Penang General Hospital, George Town 10990, Malaysia; william_olk@hotmail.com; 10Department of Urology, University Hospitals Southampton, NHS Trust, Southampton SO16 6YD, UK; bhaskarsomani@yahoo.com; 11Department of Urology, Ng Teng Fong General Hospital, Singapore 609606, Singapore; vineetgaauhaar@gmail.com

**Keywords:** kidney tumor, robot-assisted partial nephrectomy, near-infrared fluorescence, indocyanine green, estimated glomerular filtration rate

## Abstract

**Simple Summary:**

Renal cell cancer, constituting the predominant type of kidney malignancy, exhibits a global annual incidence increase of 2%. In the management of clinical T1 renal tumors, the recommended surgical approach is partial nephrectomy (PN). The primary objective of this procedure is the preservation of nephrons, thereby mitigating the risk of enduring kidney dysfunction and maintaining comparable oncological outcomes when contrasted with radical nephrectomy. Initially performed as an open surgery, PN has undergone significant evolution, with minimally invasive techniques, particularly robot-assisted ones, garnering recognition. Nonetheless, despite the inherent advantages, partial nephrectomy presents specific challenges, notably an elevated likelihood of incomplete tumor excision and positive surgical margins, which may have implications for cancer-specific outcomes. To address these challenges, the utilization of near-infrared fluorescence technology, involving the intravenous administration of indocyanine green (ICG), has been proposed. ICG, a water-soluble compound that binds to plasma albumin, emits near-infrared light, enabling the visualization of kidney vasculature and renal tumors.

**Abstract:**

Background: We aimed to analyze the influence of near-infrared fluorescence (NIRF) using indocyanine green (ICG) with standard robot-assisted partial nephrectomy (RAPN) in patients with a kidney tumor (KT). Methods: We performed a literature search on 12 September 2023 through PubMed, EMBASE, and Scopus. The analysis included observational studies that examined the perioperative and long-term outcomes of patients with a KT who underwent RAPN with NIRF. Results: Overall, eight prospective studies, involving 535 patients, were eligible for this meta-analysis, with 212 participants in the ICG group and 323 in the No ICG group. For warm ischemia time, the ICG group showed a lower duration (weighted Mean difference (WMD) = −2.05, 95% confidence interval (CI) = −3.30–−0.80, *p* = 0.011). The postoperative eGFR also favored the ICG group (WMD = 7.67, 95% CI = 2.88–12.46, *p* = 0.002). No difference emerged for the other perioperative outcomes between the two groups. In terms of oncological radicality, the positive surgical margins and tumor recurrence rates were similar among the two groups. Conclusions: Our meta-analysis showed that NIRF with ICG during RAPN yields a favorable impact on functional outcomes, whereas it exerts no such influence on oncological aspects. Therefore, NIRF should be adopted when preserving nephron function is a paramount concern.

## 1. Introduction

Renal cell cancer represents the most common kidney malignancy, and its incidence, of 2% worldwide, is increasing annually [1]. Partial nephrectomy (PN) is the recommended surgical approach for managing clinical T1 renal tumors [2]. Its primary goal is to preserve nephrons, thus mitigating the risk of long-term kidney dysfunction while potentially improving survival rates, all without compromising oncological outcomes compared to radical nephrectomy [3]. Traditionally conducted as an open surgery, PN has evolved significantly, with minimally invasive techniques now prevailing. Among the latter, robot-assisted partial nephrectomy (RAPN) has gained recognition, owing to its ability to better preserve renal function, reduce hospital stays, and offer a relatively smoother learning curve compared to laparoscopic partial nephrectomy (LPN) [4,5].

Despite the advantages it offers, partial nephrectomy presents inherent challenges, notably a heightened risk of incomplete tumor resection and positive surgical margins, which could influence cancer-specific outcomes [6]. To address these challenges, the application of near-infrared fluorescence (NIRF) technology, utilizing the intravenous administration of indocyanine green (ICG), has been proposed previously [7]. ICG, a water-soluble molecule binding to plasma albumin, emits light in the near-infrared spectrum, enabling the visualization of kidney vasculature and renal tumors [8]. Furthermore, ICG exhibits an affinity for transmembrane proteins highly expressed in proximal and distal renal tubules, rendering renal tumors less fluorescent [9]. Since 2011, NIRF technology has been effectively integrated with one of the robotic platforms, permitting surgeons to seamlessly transition between standard white light and near-infrared illumination within the same console display [10]. ICG assumes a multifaceted role in renal surgery, facilitating not only the verification of boundaries within ischemic renal regions but also the evaluation of reperfusion dynamics following unclamping and renorrhaphy procedures at the end of the surgery [11].

This study aims to perform a systematic review of the existing evidence concerning the impact of NIRF with ICG in robotic partial nephrectomy and analyze the perioperative outcomes of the use of ICG during RAPN for localized RCC.

## 2. Materials and Methods

### 2.1. Literature Search

This systematic review was performed according to the 2020 Preferred Reporting Items for Systematic Reviews and Meta-Analyses (PRISMA) method.

A literature search was performed on 12 September 2023 using PubMed, EMBASE, and Scopus, with no date limit. The following term and Boolean operators were used: (green indocyanine fluorescence OR fluorescence imaging OR near-infrared fluorescence imaging OR image-guided surgery) AND (kidney OR renal) AND (cancer OR tumor OR neoplasm) AND (partial nephrectomy OR nephron sparing surgery). The review protocol was registered in PROSPERO with the registration number (CRD42023466105).

### 2.2. Selection Criteria

The PICOS (Patient, Intervention, Comparison, Outcome, Study type) model was used to frame and answer the clinical question: P: adult patients with renal tumor; I: robot-assisted partial nephrectomy with intra-operative use of near-infrared fluorescence imaging with indocyanine green. C: standard robot-assisted partial nephrectomy; O: warm ischemia time, surgical time, console time, intraoperative blood loss, positive surgical margin, overall perioperative complications, postoperative estimated glomerular filtration rate (eGFR), blood transfusion, and postoperative stay. S: retrospective, prospective, and randomized studies.

### 2.3. Study Screening and Selection

Studies were accepted based on the PICOS eligibility criteria. Only English papers were included. Pediatric, preclinical, and animal studies were excluded. Case reports, reviews, letters to the editor, and meeting abstracts were excluded. Retrospective, prospective and prospective randomized studies were accepted. All retrieved studies were screened by two independent authors through the Covidence systematic review software, Version 2 (Veritas Health Innovation, Melbourne, Australia). A third author solved discrepancies. The full text of the screened papers was selected if it was found to be pertinent to the main outcome of this review.

### 2.4. Statistical Analysis

Meta-analyses were performed when there were two or more studies reporting the same outcomes under the same definition. The incidences of complications, positive surgical margins, blood transfusions, urinary fistulas and tumor recurrences were pooled using the Cochran–Mantel–Haenszel method with a random effect model, and were expressed as risk ratios (RR), 95% confidence intervals (CI), and *p*-values. The risk ratios of more than one (1) indicate an increased risk of complications, positive surgical margins, blood transfusions, urinary fistulas, and tumor recurrences in patients with no ICG. Warm ischemia time, surgical time, intraoperative blood loss, postoperative eGFR, and postoperative stay were pooled using the inverse variance of the weighted mean difference (WMD) with a random effect model, 95% confidence intervals, and *p*-values. Analyses were two-tailed, with a significance set at *p* < 0.05 and a 95% confidence interval. Study heterogeneity was assessed utilizing the I^2^ value. Substantial heterogeneity was defined as an I^2^ value > 50% or a chi-square *p*-value < 0.10. Meta-analysis was performed using Review Manager (RevMan) 5.4 software by Cochrane Collaboration.

## 3. Results

### 3.1. Literature Screening

The literature search retrieved 1544 papers. Duplicates, of which there were 144, were automatically excluded. We screened 1400 papers using their titles and abstracts, and 1296 papers were further rejected because they were unrelated to the aim of the present review. The remaining 92 full-text papers were evaluated for eligibility and 84 studies were excluded. Finally, eight papers were accepted and included [12,13,14,15,16,17,18,19]. Appendix A shows the flow diagram of the literature search.

### 3.2. Study Characteristics

Table 1 shows the study characteristics. Overall, there were 535 patients included in eight studies, 212 patients in the ICG group and 323 patients in the No ICG group.

Five included studies were retrospective [12,13,14,18,19], two were prospective [15,17], and one was a randomized controlled trial [16]. Five studies analyzed super-selective renal artery clamping [13,15,16,17,18], one analyzed the zero-ischemia technique [12], one analyzed both of the above [14], and one analyzed main artery clamping [19].

### 3.3. Risk of Bias Assessment

Appendix A provides a view of the quality assessment for the randomized study, revealing some concerns in the overall risk of bias. Appendix A presents the quality assessment details for the retrospective and prospective non-randomized studies. Among these, one study exhibited a critical risk of bias, two studies a serious risk of bias, while the others displayed moderate risks. The primary factors contributing to bias were the selection of participants and the bias to confounding.

### 3.4. Perioperative Outcomes

#### 3.4.1. Warm Ischemia Time

A meta-analysis of seven studies (185 cases in the ICG group and 296 cases in the No ICG group) shows that the warm ischemia time significantly favors the ICG group (WMD = −2.05, 95% CI = −3.30–−0.80, *p* = 0.011). Study heterogeneity is low (I^2^ = 51%) (Figure 1a).

#### 3.4.2. Surgical Time

A meta-analysis of six studies (140 cases in the ICG group and 251 cases in the No ICG group) shows that the surgical time shows no significant difference between the two groups (WMD = 3.79, 95% CI = −17.56–25.14, *p* = 0.73). Study heterogeneity is moderate (I^2^ = 71%) (Figure 1b).

#### 3.4.3. Intraoperative Blood Loss

A meta-analysis of eight studies (212 cases in the ICG group and 323 cases in the No ICG group) shows that the intraoperative blood loss shows no significant difference between the two groups (WMD = 1.76, 95% CI = −24.21–27.74, *p* = 0.89). Study heterogeneity is low (I^2^ = 0%) (Figure 1c).

#### 3.4.4. Postoperative Estimated Glomerular Filtration Rate

A meta-analysis of five studies (123 cases in the ICG group and 234 cases in the No ICG group) shows that the postoperative eGFR significantly favors the ICG group (WMD = 7.67, 95% CI = 2.88–12.46, *p* = 0.002). Study heterogeneity is low (I^2^ = 0%) (Figure 1d).

#### 3.4.5. Postoperative Stay

A meta-analysis of six studies (170 cases in the ICG group and 281 cases in the No ICG group) shows that the postoperative stay shows no significant difference between the two groups (WMD = −0.22, 95% CI = −0.71–0.27, *p* = 0.38). Study heterogeneity is moderate (I^2^ = 51%) (Figure 1e).

### 3.5. Postoperative Complications

#### 3.5.1. Overall Complications

A meta-analysis of eight studies (212 cases in the ICG group and 323 cases in the No ICG group) shows that overall perioperative complications show no significant difference between the two groups (RR = 0.86, 95% CI = 0.46–1.50, *p* = 0.55). Study heterogeneity is moderate (I^2^ = 48%) (Figure 2a).

#### 3.5.2. Major Complications

When stratified for the severity of complications, a meta-analysis of seven studies (185 cases in the ICG group and 296 cases in the No ICG group) shows that major perioperative complications show no significant difference between the two groups (RR = 0.77, 95% CI = 0.32–1.82, *p* = 0.55). Study heterogeneity is low (I^2^ = 0%) (Figure 2b).

#### 3.5.3. Minor Complications

Similarly, a meta-analysis of seven studies (191 cases in the ICG group and 296 cases in the No ICG group) shows that minor perioperative complications show no significant difference between the two groups (RR = 0.81, 95% CI = 0.44–1.50, *p* = 0.50). Study heterogeneity is low (I^2^ = 24%) (Figure 2c).

#### 3.5.4. Urinary Fistula Rate

Evaluating the specific postoperative complications, a meta-analysis of five studies (117 cases in the ICG group and 143 cases in the No ICG group) shows that the urinary fistula (UF) rate shows no significant difference between the two groups (RR = 2.85, 95% CI = 0.12–63.83, *p* = 0.52). Study heterogeneity is not applicable (Figure 2d).

#### 3.5.5. Blood Transfusion Rate

A meta-analysis of seven studies (191 cases in the ICG group and 217 cases in the No ICG group) shows that the blood transfusion rate shows no significant difference between the two groups (RR = 0.71, 95% CI = 0.27–1.89, *p* = 0.50). Study heterogeneity is low (I^2^ = 24%) (Figure 2e).

### 3.6. Oncological Outcomes

#### 3.6.1. Positive Surgical Margins

A meta-analysis of eight studies (212 cases in the ICG group and 323 cases in the No ICG group) shows that the positive surgical margins rate shows no significant difference between the two groups (RR = 1.16, 95% CI = 0.46–2.92, *p* = 0.76). Study heterogeneity is low (I^2^ = 7%) (Figure 3a).

#### 3.6.2. Renal Tumor Recurrence

A meta-analysis of four studies (103 cases in the ICG group and 214 cases in the No ICG group) shows that the renal tumor recurrence rate shows no significant difference between the two groups (RR = 0.69, 95% CI = 0.04–12.98, *p* = 0.81). Study heterogeneity is not applicable (Figure 3b).

## 4. Discussion

This systematic review and associated meta-analyses reveals that the utilization of indocyanine green (ICG) during RAPN is associated with a reduced warm ischemia time and an elevated postoperative eGFR, when compared to RAPN performed without NIRF guidance. Nevertheless, no statistically significant disparities were observed between the two groups in relation to various perioperative outcomes, including surgical duration, intraoperative blood loss, positive surgical margins, postoperative complications (both major and minor), and the length of the postoperative hospital stay. Furthermore, the rate of renal tumor recurrence appeared to be similar, irrespective of the application of NIRF guidance.

The optimal duration of warm ischemia during partial nephrectomy remains a topic of ongoing debate. Studies have investigated the relationship between warm ischemia time and renal function in patients with a solitary kidney who underwent partial nephrectomy with hilar clamping. A prolonged warm ischemia time was found to be associated with an increased risk of acute renal failure and the development of new-onset stage IV chronic kidney disease, with respective odds ratios of 1.05 and 1.06 for every 1 min increment [20].

Efforts have been made to minimize or eliminate renal ischemia during robot-assisted partial nephrectomy, with the most radical approach being off-clamp partial nephrectomy. Propensity score analyses have demonstrated that off-clamp partial nephrectomy can help preserve post-operative renal function, albeit with a slightly higher estimated blood loss [21]. However, a recent randomized controlled trial comparing on-versus off-clamp RAPN in patients with a normal baseline kidney function and two kidneys found no significant differences between the two strategies [22].

Large international studies have consistently shown that main renal artery clamping remains the prevailing technique in RAPN, even in cases involving chronic kidney disease or solitary kidneys [23]. Therefore, the pursuit of alternative methods to reduce warm ischemia time without increasing the risk of bleeding and transfusion is of paramount importance. Early unclamping techniques during robot-assisted laparoscopic partial nephrectomy have emerged as an attractive option for minimizing warm ischemia without raising morbidity concerns [24,25]. Another method for eliminating renal ischemia during robotic partial nephrectomy, while avoiding potential complications associated with the off-clamp technique, involves the use of NIRF imaging to facilitate super-selective arterial clamping in a ‘zero-ischemia’ approach. Moreover, patients who underwent RAPN with intravenous ICG for selective clamping exhibited significantly higher postoperative eGFR compared to those without ICG injection, safeguarding the blood supply to the healthy renal parenchyma and minimizing its damage. In a pooled analysis conducted by Veccia et al., apart from the observed reduction in WIT, the application of NIRF group demonstrated higher values of eGFR during the short-term postoperative follow-up (1–3 months) (WMD: 9.26 mL/min; 95% CI: 6.46, 12.06; *p* < 0.001), despite a similar postoperative eGFR at discharge [26]. Consequently, it is evident that the utilization of ICG may yield superior short-term renal functional outcomes.

Various tools have been implemented in minimally invasive partial nephrectomy for the precise identification of renal tumors, aiming to ensure a reduced WIT and enhanced preservation of the healthy parenchyma. Previously, the use of intraoperative ultrasound had been recommended, particularly in cases involving higher complexity masses [27]. Recently, a three-dimensional planning tool has been introduced, facilitating the selective clamping of the renal artery, and showing a high level of accuracy in kidney anatomy [28]. However, despite lower rates of detriment and surgical injury to the kidney associated with this model, significant benefits in oncological or functional outcomes are not yet certain [29].

In the near-infrared fluorescence mode, the renal tumor exhibits a distinct color shade compared to the healthy parenchyma. However, based on our experience, this difference does not appear to enhance oncological outcomes. The fundamental goal of conservative surgery is to maximize the preservation of healthy tissue while completely removing the tumor, avoiding any residual tumor tissue at the resection margins. Our meta-analysis indicates a similar number of positive surgical margins between standard RAPN and ICG-assisted RAPN, with a total of nine cases of positive margins recorded in all ICG group studies. This underscores the need for further refinements in this technique to achieve greater oncological precision in tumor enucleation.

RAPN may be associated with a diverse spectrum of postoperative complications, encompassing vascular complications, UF, and injury to surrounding structures. In the former, hemorrhage often arises from inadequate vascular control, frequently due to the missed segmental vessel during selective or super-selective clamping, or improper securing of hemostasis after tumor excision, occurring in approximately 1.6–8.6% of patients [30]. In our meta-analysis, no significant difference in intraoperative blood loss or blood transfusion rate was observed. These outcomes appeared consistent regardless of tumor complexity or the type of ischemia technique employed in the included studies in this MA (Meta-analysis). Despite the theoretical advantages of ICG, such as providing additional information on tissue perfusion for precise clamping and delineating renal parenchyma for meticulous dissection [31], our analysis suggests that ICG does not confer an added advantage in minimizing intraoperative bleeding during RAPN. We must note that we lack sufficient data to assess the utility of ICG in preventing subsequent renal arteriovenous fistula and other vascular complications. Similarly, our analysis did not reveal a significant difference in UF rate between the two cohorts (RR = 2.85, 95% CI = 0.12–63.83, *p* = 0.52). The incidence of UF after RAPN is relatively low, at <5% in our patient cohort, consistent with existing literature [32], compared to 2–6% following open and laparoscopic procedures [33,34]. Tumor size, blood loss, and ischemia time are significantly associated with UF development [32]. Although our meta-analysis shows a significant advantage in warm ischemia time favoring the ICG group (WMD = −2.05, 95% CI = −3.30–−0.80, *p* = 0.011), this advantage did not influence UF occurrence.

With regard to injury to surrounding structures, numerous organs may be susceptible to harm during RAPN, including bowel injury (0.3–0.5%), pleural injury (0.6–12.9%), splenic injury (0.5–4.3%), hepatobiliary injury (0.1–1.4%), pancreatic injury (0.2–2.1%), and lymphatic injuries [30,32,35]. Klassen et al. noted a limitation in using ICG, where toggling between white light and near-infrared fluorescence imaging resulted in a dark intracorporeal field relative to the ICG-illuminated kidney, possibly increasing the risk of iatrogenic injury to surrounding structures by the robotic surgeon [36]. Additionally, the green color of ICG dye may cause margin clarity issues during mass excision. However, our meta-analysis, comprising seven studies (185 cases with ICG and 296 cases without ICG), indicates no significant difference in major perioperative complications related to organ injury between the two groups (RR = 0.77, 95% CI = 0.32–1.82, *p* = 0.55). This may be attributed to the enhanced precision control provided by the robotic system in experienced hands, rendering it less reliant on ICG for improved tumor margin definition.

Regarding the prevention of complications, NIRF using ICG has been adopted as a practical tool in open and laparoscopic partial nephrectomies for identifying anatomical structures, aiding in devascularization verification, indicating resection margins, and enhancing surgical planning [37]. However, its value in RAPN does not appear to provide additional benefits based on our MA findings.

The principal objective of PN is to achieve oncological radicality, comparable to that of radical nephrectomy, while enhancing functional outcomes with respect to renal function preservation [2]. Preoperative renal function, warm ischemia time, and the volume of preserved parenchyma have been identified as pivotal predictive factors for preserving preoperative renal function [18]. Three primary factors contribute to the improvement in eGFR preservation when employing ICG during RAPN:

(a)A reduced warm ischemia time is achieved through super-selective clamping, as ICG enables precise identification of the arterial blood supply to the tumor [38].(b)There is enhanced identification of the tumor’s location and margins due to differential fluorescence characteristics between the mass and normal parenchyma, allowing for the continuous marking of the lesion throughout the procedure [39].(c)There can be direct evaluation of the ischemic effect on surrounding parenchyma after renorrhaphy, which is another contributing factor to postoperative eGFR loss, and this evaluation can be accomplished with a conventional intravenous injection of ICG to confirm the absence of ischemic injury to healthy parenchyma [38,39].

In a retrospective analysis by Yang et al. [19], comparing 111 RAPN cases with 21 ICG–RAPN cases, early improved eGFR preservation was observed in the ICG group. However, this trend was not sustained at the three-month follow-up, suggesting possible compensation of the contralateral kidney and nephron recovery in the non-ICG group.

Evaluating the follow-up, our analysis did not show any significant difference between IGC–RAPN and standard RAPN in tumor recurrence rates. Although Yang et al. [19] reported a few cases of tumor recurrence in the non-ICG group, no statistical correlation was established. Therefore, the use of ICG in RAPN has demonstrated excellent oncological outcomes with minimal recurrence rates, but it has not yet proven its superiority over standard RAPN. Although ICG enhances the visualization of tumor margins, particularly for irregular outlines, this advantage does not translate into a noticeable difference in the rate of positive surgical margins, as indicated by our meta-analysis. Furthermore, according to Simone et al. [39], the presence of focal positive surgical margins does not necessarily correlate with long-term recurrences. Long-term recurrences are more associated with factors such as histology, staging, and grade. While ICG has proven valuable in identifying tumor sites and vascularization, our data suggest that the high-resolution 3D imaging provided by the robotic system, combined with precise dissection, remain the primary determinants of functional outcomes in RAPN. Given the limited current publications in this area, further studies are warranted to investigate these features in greater detail.

We have conducted a comprehensive meta-analysis, synthesizing a diverse body of research studies that collectively provide substantial evidence supporting the intraoperative use of ICG during partial nephrectomy. Our findings unequivocally endorse the adoption of ICG before tumor resection in the context of warm ischemia time and postoperative eGFR, as long as this technique can maximize the preservation of healthy renal parenchyma while minimizing hypoxic damage. Consequently, the clinical scenarios in which nephron-sparing surgery is imperative, such as in cases of solitary kidney, multiple renal masses, or chronic renal insufficiency, should also warrant the recommendation for the incorporation of indocyanine green.

Conversely, oncological radicality remains unaffected, as demonstrated by positive surgical margins and tumor recurrence. Indeed, the consistent application of an extremely thin layer of healthy renal tissue during robotic tumor enucleation consistently achieves negative surgical margins in the majority of patients, even when facing complete invasion of the pseudocapsule [7].

Similarly, perioperative outcomes, including intraoperative blood loss, postoperative complications, and postoperative hospital stay, were found to be independent of the use of indocyanine green. Therefore, the inclusion of ICG does not necessarily guarantee an improved safety profile.

Nevertheless, the current study is not devoid of limitations. Firstly, the majority of the included papers are retrospective in nature, making them susceptible to inherent biases. Secondly, there is significant heterogeneity in the approach to renal artery clamping across the studies. Indeed, some studies employ a clampless technique, while others assess super-selective clamping, and in yet others, the main renal artery is clamped. Thirdly, it should be noted that in several studies, the nephrometry score was recorded as a nominal variable rather than a continuous one. This approach thus precluded the ability to make a meaningful comparison of the average complexity of renal masses between the two groups.

## 5. Conclusions

Our meta-analysis comprehensively evaluates the various facets of RAPN and NIRF. Perioperatively, NIRF may aid in the precise delineation of non-tumor margins. However, this does not necessarily translate into improved oncological outcomes, nor does it reduce blood loss or perioperative complications. Nevertheless, from a functional standpoint, NIRF significantly contributes to achieving a more accurate representation of renal functional status through better measurement of eGFR. This functional enhancement is particularly valuable in intricate surgeries that necessitate maximum renal preservation.

There is insufficient evidence to suggest that NIRF-assisted surgery can effectively minimize recurrence or post-operative margins. Consequently, the therapeutic oncological potential of NIRF in the context of RAPN remains uncertain.

## Figures and Tables

**Figure 1 cancers-15-05560-f001:**
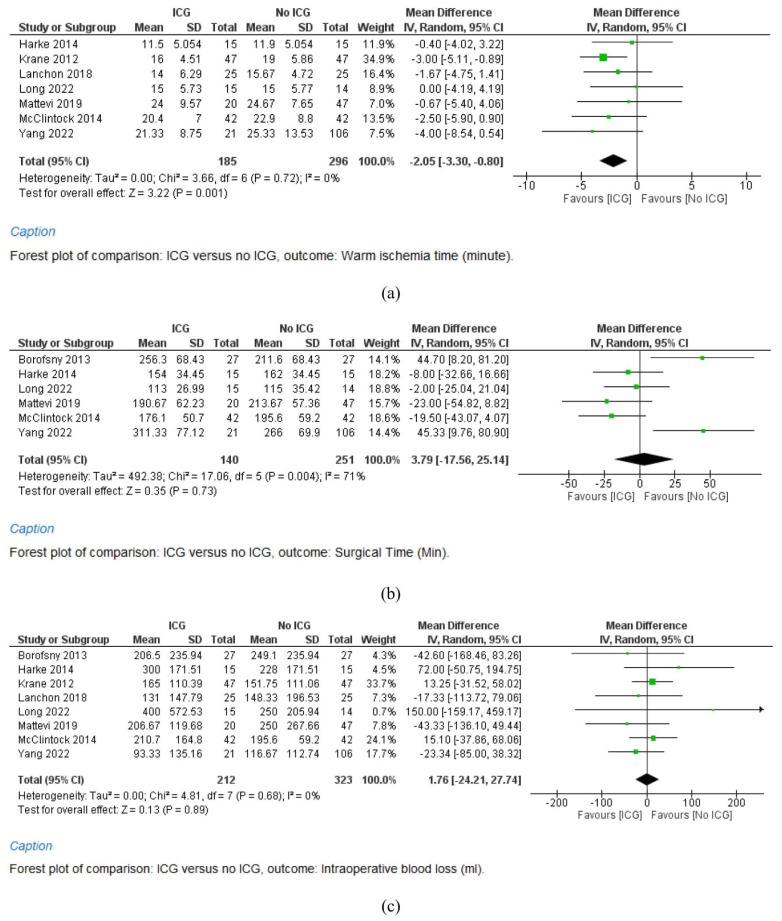
Studies concerning the warm ischemia time (**a**), surgical time (**b**), intraoperative blood loss (**c**), estimated glomerular filtration rate (eGFR) (**d**), and postoperative stay (**e**) during robot-assisted partial nephrectomy (RAPN) with and without indocyanine green (ICG) use.

**Figure 2 cancers-15-05560-f002:**
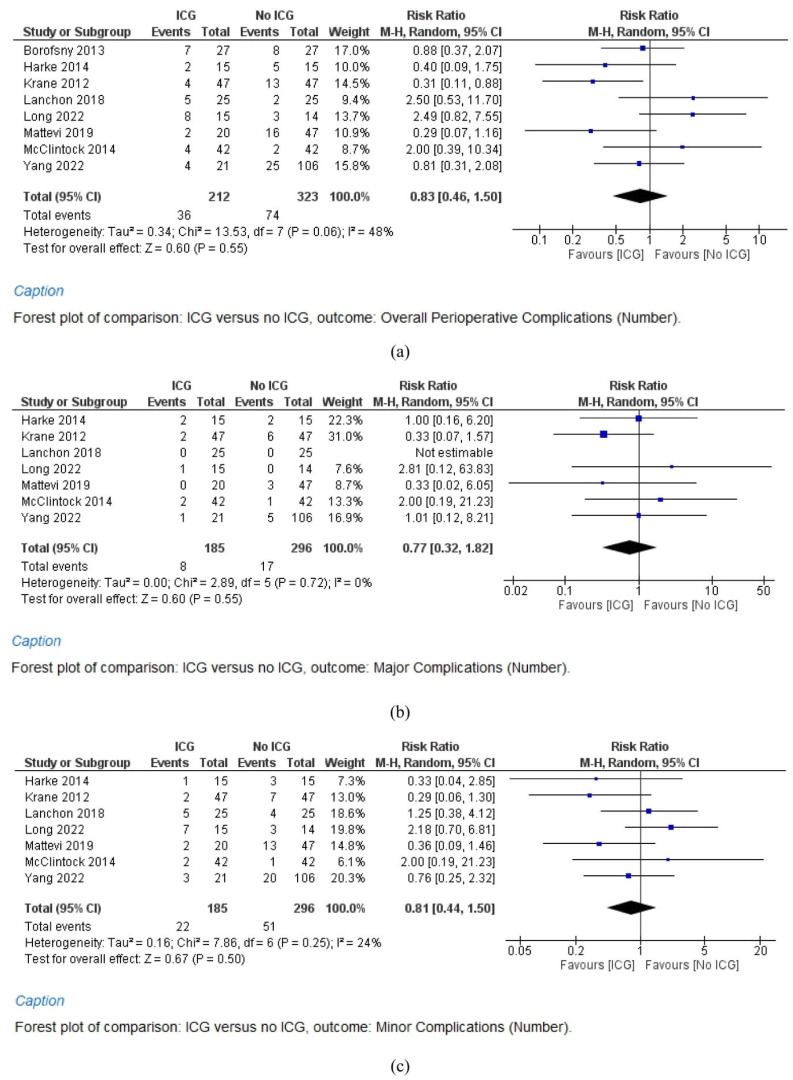
Studies concerning the overall complications (**a**), major complications (**b**), minor complications (**c**), urinary fistula (**d**), and blood transfusion rates (**e**) after robot-assisted partial nephrectomy (RAPN) with and without indocyanine green (ICG) use.

**Figure 3 cancers-15-05560-f003:**
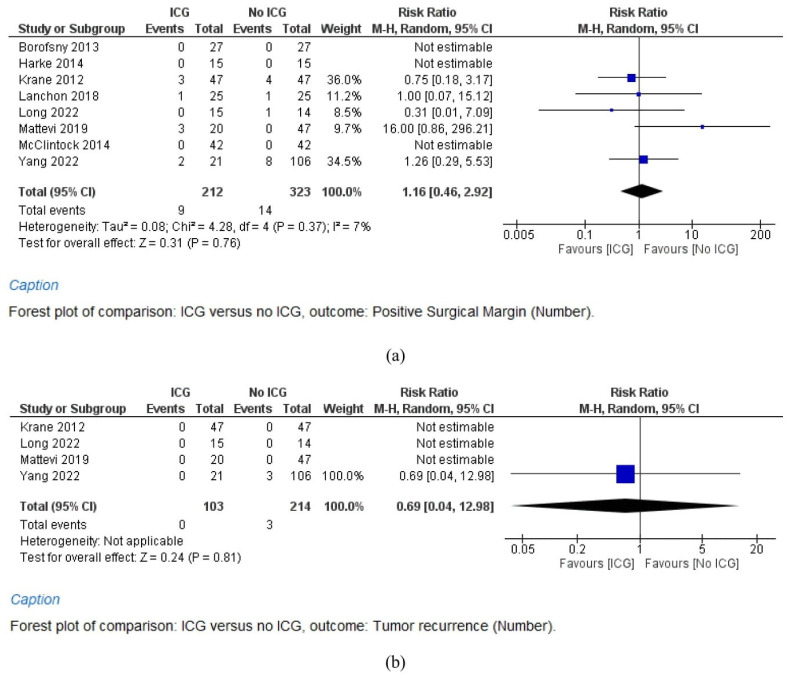
Studies concerning the positive surgical margin rate (**a**), and tumor recurrence (**b**) rates after robot-assisted partial nephrectomy (RAPN) with and without indocyanine green (ICG) use.

**Table 1 cancers-15-05560-t001:** Studies comparing near-infrared fluorescence (NIRF) using indocyanine green (ICG) with standard robot-assisted partial nephrectomy (RAPN) in patients with a kidney tumor. SSC: super-selective clamping; ZI: zero-ischemia; MAC: main artery clamping; eGFR: estimated glomerular filtration rate.

First Author, Year	Type of Study	Number of Cases	Mean Age, Years	Clamping Technique	Surgical Time, min	Positive Surgical Margin, *n* (%)	Overall Complications, *n* (%)	Postoperative eGFR	Number of Control	Mean Age, Years	Clamping Technique	Surgical Time, min	Positive Surgical Margin, *n* (%)	Overall Complications, *n* (%)	Postoperative eGFR	Final Considerations
Borofsny (2013) [12]	Retrospective	27	60	ZI	256.3	0	7 (26)	70.8	27	58.1	MAC	211.6	0	8 (27)	-	ZI partial nephrectomy is feasible most of time with NIRF, with a greater renal function preservation.
Harke (2014) [13]	Retrospective	15	63	SSC	154	0	2 (13)	-	15	63.2	MAC	162	0	5 (33)	-	NIRF can be employed safely, especially in hilar and intrarenal as well as polar tumors, avoiding ischemic injury to the remaining parenchyma, with superior kidney function preservation
Krane (2012) [14]	Retrospective	47	60	SSC or ZI	-	3 (6)	4 (9)	-	47	60.2	SSC or ZI	-	4 (9)	13 (28)	-	RPN using NIRF–ICG can be performed safely and effectively. Differential ICG uptake by different tumors did not lead to significant differences in the positive margin rate.
Lanchon (2018) [15]	Prospective	25	66	SSC	-	1 (4)	5 (20)	78	25	68	MAC	119	1 (4)	4 (16)	72.67	Super-selective clamping with NIRF using ICG is safe and feasible, leading to an increased preservation of overall and split postoperative renal function, while keeping the benefit of main artery clamping on blood loss and perioperative complications.
Long (2022) [16]	Randomized Controlled Trial	15	56	SSC	113	0	8 (53)	80	14	61	MAC	115	1 (7)	3 (21)	79.83	SSC–RAPN using NIRF did not provide better renal preservation than renal artery clamping in non-selected patients,.
Mattevi (2019) [17]	Prospective	20	61	SSC	190.67	3 (7.1)	2 (10)	46.83	47	66	MAC	213.67	0	18 (38)	38.67	There was an association between NIRF and improved short-term functional outcomes, as measured by eGFR at renal scan.
McClintock (2014) [18]	Retrospective	42	59	SSC	176.1	0	4 (10)	78.2	42	59.4	MAC	195.6	0	2 (5)	68.5	The use of NIRF aids in the implementation of selective arterial clamping, enabling real-time verification of the intended ischemic regions, with enhanced short-term functional outcomes.
Yang (2022) [19]	Retrospective	21	58	MAC	311.33	2 (10)	4 (19)	79.6	106	57	MAC	266	8 (8)	25 (23)	71.33	ICG determined superior short-term renal functional outcomes, with less operative blood loss. Therefore, ICG–RAPN is an ostensibly safe procedure with potentially superior short-term renal functional outcomes.

## Data Availability

The data will be provided by the corresponding author upon a reasonable request.

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
