# Peer review of "Near-Infrared Fluorescence Imaging with Indocyanine Green for Robot-Assisted Partial Nephrectomy: A Systematic Review and Meta-Analysis"

_cancers, 2023, doi:10.3390/cancers15235560_

Round 1

Reviewer 1 Report

Comments and Suggestions for Authors

The paper is a review and deals with renal cell cancer.
As the authors write audibly, it is the most common malignant neoplasm of the kidney. Therefore, the subject is fully justified and such reviews provide a lot of interesting information. Partial nephrectomy is obviously more beneficial to the patient than radical nephrectomy, but of course it must meet certain criteria.
The introduction is brief but sufficient. Materials and methods describe how the literature was selected with exclusions. The criteria used resulted in the qualification of only 8 papers out of an initial search of 1544.
As results, the authors describe and analyze the results on specific issues such as the duration of surgery or margin preservation (3.4.1.- 3.4.9.).
The posted ilutations are not very readable. I suggest that Figure1. and Figure 2. should be significantly enlarged, and perhaps even a separate page should be devoted to each.
The discussion is conducted in a proper way.
References are appropriate.

Author Response

REVIEWER #1 COMMENTS

The paper is a review and deals with renal cell cancer.

As the authors write audibly, it is the most common malignant neoplasm of the kidney. Therefore, the subject is fully justified and such reviews provide a lot of interesting information. Partial nephrectomy is obviously more beneficial to the patient than radical nephrectomy, but of course it must meet certain criteria.

The introduction is brief but sufficient. Materials and methods describe how the literature was selected with exclusions. The criteria used resulted in the qualification of only 8 papers out of an initial search of 1544.

REPLY. Thank you for your nice words.

As results, the authors describe and analyze the results on specific issues such as the duration of surgery or margin preservation (3.4.1.- 3.4.9.).

The posted ilutations are not very readable. I suggest that Figure1. and Figure 2. should be significantly enlarged, and perhaps even a separate page should be devoted to each.

REPLY. Thank you for the comment. As you suggested, We enlarged the figures and reorganized them in light of the modifications made in the results section.

The discussion is conducted in a proper way.

References are appropriate.

REPLY. Thank you for your nice words.

Reviewer 2 Report

Comments and Suggestions for Authors

Dear Authors,

I read the paper “Near-Infrared Fluorescence Imaging with Indocyanine Green for Robot-Assisted Partial Nephrectomy: A systematic review and meta-analysis” with interest.

The aim of the study was to comprehensively review and meta-analyzed available comparative studies about the use of ICG during RAPN. The systematic review follows a rigorous process and the methodology is strong, Authors should be commended for their efforts.

Overall, the paper is interesting. Few remarks:

Title: adequate

Abstract: adequate

Introduction: adequate. I would specify that another application of ICG is during selective clamping  to double check the boundaries of ischemic renal regions. Furthermore, it can be used to assess the vascularization of the sutured parenchyma at the end of RAPN. This paper might help you in improving this section. 10.23736/S2724-6051.23.05161-3

Methods: adequate

Results:

-       I believe it would be better for readers to present the results in two main section (for example: intra- and postoperative results) and focusing more on significant findings, rather than presenting all the analysis. Please consider reorganize this section.

Discussion:

-       The use 3D virtual models is pushing towards a larger adoption of selective/super-selective clamping strategy and enucleation technique during minimally invasive PN, as also demonstrated by this recent SR 10.1016/j.euo.2022.09.003.

These advantages would be even more evident with the adoption of new generation 3D models as demonstrate in these recent experiences (10.1016/j.eururo.2023.01.003)

Finally, a SR on this topic was already published in 2018 by Veccia et al. (with almost the same number of included studies). I would recommend to compare their findings with your results and highlight pros and cons of your work.

Author Response

Dear Authors,

I read the paper “Near-Infrared Fluorescence Imaging with Indocyanine Green for Robot-Assisted Partial Nephrectomy: A systematic review and meta-analysis” with interest.

The aim of the study was to comprehensively review and meta-analyzed available comparative studies about the use of ICG during RAPN. The systematic review follows a rigorous process and the methodology is strong, Authors should be commended for their efforts.

Overall, the paper is interesting.

REPLY. Thank you for your nice words.

Few remarks:

Title: adequate

REPLY. Thank you for your nice words.

Abstract: adequate

REPLY. Thank you for your nice words.

Introduction: adequate. I would specify that another application of ICG is during selective clamping to double check the boundaries of ischemic renal regions. Furthermore, it can be used to assess the vascularization of the sutured parenchyma at the end of RAPN. This paper might help you in improving this section. 10.23736/S2724-6051.23.05161-3

REPLY. Thank you for your comment. In accordance with your suggestion, we have incorporated a statement highlighting additional roles of the ICG in the introduction section, as follows “ICG assumes a multifaceted role in renal surgery, facilitating not only the verification of boundaries within ischemic renal regions but also the evaluation of reperfusion dynamics following unclamping and renorrhaphy procedures at the end of the surgery [11].

Methods: adequate

REPLY. Thank you for your nice words.

Results:

-       I believe it would be better for readers to present the results in two main section (for example: intra- and postoperative results) and focusing more on significant findings, rather than presenting all the analysis. Please consider reorganize this section.

REPLY. Thank you for your comment. As per your suggestion, we have decided to reorganize our results and categorize them into three sections (perioperative outcomes, postoperative complications, and oncological outcomes), as follows: “3.4. Perioperative outcomes

3.4.1. Warm Ischemia Time

Meta-analysis from seven studies (185 cases in ICG and 296 cases in No ICG) shows that the warm ischemia time significantly favors the ICG group (WMD -2.05, 95% CI -3.30– -0.80, p=0.011). Study heterogeneity is low (I2 51%) (Figure 1a).

3.4.2. Surgical Time

Meta-analysis from six studies (140 cases in ICG and 251 cases in No ICG) shows that the surgical time shows no significant difference between the two groups (WMD 3.79, 95% CI -17-56– 25.14, p=0.73). Study heterogeneity is moderate (I2 71%) (Figure 1b).

3.4.3. Intraoperative blood loss

Meta-analysis from eight studies (212 cases in ICG and 323 cases in No ICG) shows that the intraoperative blood loss shows no significant difference between the two groups (WMD 1.76, 95% CI -24.21– 27.74, p=0.89). Study heterogeneity is low (I2 0%) (Figure 1c).

3.4.4. Postoperative estimated Glomerular Filtration Rate

Meta-analysis from five studies (123 cases in ICG and 234 cases in No ICG) shows that the postoperative eGFR significantly favors the ICG group (WMD 7.67, 95% CI 2.88– 12.46, p=0.002). Study heterogeneity is low (I2 0%) (Figure 1d).

3.4.5. Postoperative stay

Meta-analysis from six studies (170 cases in ICG and 281 cases in No ICG) shows that the postoperative stay shows no significant difference between the two groups (WMD -0.22, 95% CI -00.71– 0.27, p=0.38). Study heterogeneity is moderate (I2 51%) (Figure 1e).

3.5 Postoperative complications

3.5.1. Overall Complications

Meta-analysis from eight studies (212 cases in ICG and 323 cases in No ICG) shows that the overall perioperative complication shows no significant difference between the two groups (RR 0.86, 95% CI 0.46– 1.50, p=0.55). Study heterogeneity is moderate (I2 48%) (Figure 2a).

3.5.2. Major Complications

When stratified for the severity of complications, meta-analysis from seven studies (185 cases in ICG and 296 cases in No ICG) shows that the major perioperative complica-tion shows no significant difference between the two groups (RR 0.77, 95% CI 0.32– 1.82, p=0.55). Study heterogeneity is low (I2 0%) (Figure 2b).

3.5.3. Minor Complications

Similarly, meta-analysis from seven studies (191 cases in ICG and 296 cases in No ICG) shows that the minor perioperative complication shows no significant difference between the two groups (RR 0.81, 95% CI 0.44– 1.50, p=0.50). Study heterogeneity is low (I2 24%) (Figure 2c). 

3.5.4.  Urinary fistula rate

Evaluating the specific postoperative complications, meta-analysis from five studies (117 cases in ICG and 143 cases in No ICG) shows that the urinary fistula rate shows no significant difference between the two groups (RR 2.85, 95% CI 0.12– 63.83, p=0.52). Study heterogeneity is not applicable (Figure 2d).

3.5.5. Blood transfusion rate

Meta-analysis from seven studies (191 cases in ICG and 217 cases in No ICG) shows that the blood transfusion rate shows no significant difference between the two groups (RR 0.71, 95% CI 0.27– 1.89, p=0.50). Study heterogeneity is low (I2 24%) (Figure 2e).

3.6 Oncological outcomes

3.6.1. Positive Surgical Margins

Meta-analysis from eight studies (212 cases in ICG and 323 cases in No ICG) shows that the positive surgical margins rate shows no significant difference between the two groups (RR 1.16, 95% CI 0.46– 2.92, p=0.76). Study heterogeneity is low (I2 7%) (Figure 3a).

3.6.2 Renal tumor recurrence

Meta-analysis from four studies (103 cases in ICG and 214 cases in No ICG) shows that the renal tumor recurrence rate shows no significant difference between the two groups (RR 0.69, 95% CI 0.04– 12.98, p=0.81). Study heterogeneity is not applicable (Figure 3b).”.

Discussion:

-       The use 3D virtual models is pushing towards a larger adoption of selective/super-selective clamping strategy and enucleation technique during minimally invasive PN, as also demonstrated by this recent SR 10.1016/j.euo.2022.09.003.

These advantages would be even more evident with the adoption of new generation 3D models as demonstrate in these recent experiences (10.1016/j.eururo.2023.01.003)

REPLY. Thank you for your comment. As you suggested we implemented the topic on 3D model in the discussion section, as follows: “ Various tools have been implemented in minimally invasive partial nephrectomy for the precise identification of renal tumors, aiming to ensure a reduced WIT and enhanced preservation of the healthy parenchyma. Previously, the use of intraoperative ultrasound had been recommended, particularly in cases involving higher complexity masses [27]. Recently, a Three-dimensional Planning Tool has been introduced, facilitating the selective clamping of the renal artery, and showing a high level of accuracy of kidney anatomy [28]. However, despite lower rates of detriment and surgical injury to the kidney associated with this model, significant benefits in oncological or functional outcomes are not yet certain [29].”

Finally, a SR on this topic was already published in 2018 by Veccia et al. (with almost the same number of included studies). I would recommend to compare their findings with your results and highlight pros and cons of your work.

REPLY. Thank you for your comment. As you suggested we implemented this pooled analysis, as follows: “In a pooled analysis conducted by Veccia et al., apart from the observed reduction in WIT, the application of NIRF demonstrated higher values of eGFR during the short-term post-operative follow-up (1–3 months) (WMD: 9.26 ml/min; 95% CI: 6.46, 12.06; p < 0.001), de-spite a similar postoperative eGFR at discharge [26]. Consequently, it is evident that the utilization of ICG may yield superior short-term renal functional outcomes.”

Round 2

Reviewer 2 Report

Comments and Suggestions for Authors

I think thta the paper has been improved.